# Synthesis and Dielectric Relaxation Studies of $K_xFe_yTi_{8-y}O_{16}$ (x = 1.4–1.8 and y = 1.4–1.6) Ceramics with Hollandite Structure

**Alexey Tsyganov** [1,*], **Denis Artyukhov** [2], **Maria Vikulova** [1], **Natalia Morozova** [1], **Ilya Zotov** [1], **Sergey Brudnik** [1], **Aleksandra Asmolova** [1], **Denis Zheleznov** [1], **Alexander Gorokhovsky** [1] and **Nikolay Gorshkov** [1,*]

[1] Department of Chemistry and Technology of Materials, Yuri Gagarin State Technical University of Saratov, 77 Polytecnicheskaya Street, 410054 Saratov, Russia

[2] Department of Power and Electrical Engineering, Yuri Gagarin State Technical University of Saratov, 77 Polytecnicheskaya Street, 410054 Saratov, Russia

[*] Correspondence: tsyganov.a.93@mail.ru (A.T.); gorshkov.sstu@gmail.com (N.G.)

**Abstract:** Some solid solutions with the chemical composition $K_xFe_yTi_{8-y}O_{16}$ (KFTO) and a hollandite-like structure were successfully synthesized by modified sol–gel method. The obtained powders were characterized using X-ray diffraction (XRD) and scanning electron microscopy (SEM). The ceramic pellets based on KFTO powders were obtained by compressing and sintering at 1080 °C for 4 h. The sinters were characterized by X-ray and impedance spectroscopy. XRD results show that KFTO powders have a mono-phase tetragonal structure at x = 1.4–1.8 and y = 1.4–1.6. However, it was recognized that the hollandite-like phase could be broken during sintering to form $TiO_2$ and $Fe_2TiO_5$ crystals distributed throughout the volume of the ceramics. A frequency dependency of dielectric properties for the sinters was studied by impedance spectroscopy. It was found that an increase in the $TiO_2$ (rutile) phase during the sintering contributes to a decrease in dielectric losses. At the same time, the KFTO ceramics with reduced content of potassium had increased permittivity. The contribution of electron-pinned defect dipoles (EPDD) and internal barrier layer capacitance (IBLC) in the permittivity of the obtained ceramics was estimated using the Havriliak–Negami equation. It is shown that the KFTO ceramics have the polydisperse characteristic of dielectric relaxation. The observed grain and grain boundary dipole relaxation times were $1.03 \times 10^{-6}$ to $5.51 \times 10^{-6}$ s and 0.197 to 0.687 s, respectively.

**Keywords:** ceramics; hollandite; permittivity; internal barrier layer capacitance (IBLC); electron-pinned defect dipoles (EPDD); Havriliak–Negami equation



## 1. Introduction

Nowadays, complex oxides with a layered and tunnel structure have attracted much attention in terms of their synthesis, modification, and control of various chemical and physical functional properties.

The hollandite-type compounds became the most popular due to the variety of their chemical compositions and structural features. A general chemical formula of hollandites can be described as $A_xB_8O_{16}$, where x ≤ 2, A is a mono- or two-valent ion (i.e., Na, Ag, K, Tl, Rb, Cs, Ba, Sr, and Pb) and B is any ion, with a valence varied in the range from two to five, forming octahedral sites (i.e., Mg, Cu, Zn, Co, Ni, Al, Ga, Fe, In, Sc, Cr, Si, Ge, Ti, Mn, Sn, and Sb). The ideal hollandite-like structure has a tetragonal symmetry, I4/m, and is formed by $BO_6$ octahedra in the form of two types of 1D channels (1 × 1 and 2 × 2 octahedra wide), and the A-ions occupy 2 × 2 octahedra-wide channels [1–3]. Consequently, there are several applications of oxides with the hollandite-like structure: (I) as a host for the immobilization of radioactive wastes [4–7]; (II) as ion sieves [8–10]; (III) as cathode materials for lithium or potassium batteries [11–14]; and (IV) as conductors for alkali ions [15–17].

The $A_x(Ti^{4+}, Ti^{3+})_8O_{16}$ titanates constitute a particular group of complex oxides with a hollandite-type structure. These substances are of great interest due to a wide range of possibilities to modify their structure and properties to improve functional properties [18]. Various two- and three-valent transition metal cations can be easily introduced into the hollandite-type framework by $Ti^{3+}$ replacing, and transformed in the solid solutions $A_x[Ti^{4+}, M^{3+/2+}]_8O_{16}$ (M is $Al^{3+}$, $Fe^{3+}$, $Mg^{2+}$, $Ti^{3+}$, $Ga^{3+}$, $Cr^{3+}$, $Sc^{3+}$, etc.) [19–24].

Most of the scientific research on Fe-doped hollandites, including $K_{1.46}Fe_{0.8}Ti_{7.2}O_{16}$ [25] and $K_{1.75}Fe_{1.75}Ti_{6.25}O_{16}$ [26], is devoted to the development of synthesis procedures that allow for influencing the chemical composition, structure, and properties of the final materials, as well as a stability of the hollandite-like structure [27–29].

There are several works on investigations of the electrical properties of titanate hollandites. In particular, the electrical conductivity of some hollandites, obtained in the system of $K_2O$–$MeO$–$TiO_2$ (Me = Mg, Ni, Cu) by sol–gel synthesis in the air and $H_2$ atmosphere, was investigated. The obtained results show that these materials are promising in the in the production of gas sensors and electrode materials in the hydrogen fuel cells [30]. In the research [31], it was shown that the titanate hollandites such as $Ba_{1.33}Ga_{2.67}Ti_{5.33}O_{16}$, $K_{1.33}Ga_{1.33}Ti_{6.67}O_{16}$, and $K_{1.54}Mg_{0.77}Ti_{7.23}O_{16}$ have high electron conductivity at high temperatures in the $H_2$ atmosphere. In the work [32], the electrical conductivity of the ceramics obtained in the $K_2O$–$TiO_2$–$Fe_2O_3$ system were also studied, but a comprehensive study of the dielectric properties of iron-doped titanate hollandites has not been carried out yet.

In this regard, the complex investigation of permittivity, dielectric losses, and conductivity of new kinds of the hollandite-like solid solutions remains an urgent task. In recent works, it was noted that $K_x(Me,Ti)_8O_{16}$ materials, which can be synthesized by solid-state, sol–gel, and hydrothermal methods [25,33], had high permittivity [34–36]. In particular, the ceramics obtained by calcination of $K_{1.6}(Cu_{0.8}Ti_{7.2})O_{16}$ powder at T > 1050 °C had $\varepsilon$ ~$10^4$ and tan $\delta$ ~0.2–0.6 at low frequencies. In addition, hollandite-like solid solutions can be used as effective ceramic fillers in the dielectric polymer matrix composites due to their stable and high permittivity [37,38]. However, the mechanism, describing high permittivity of hollandite-like solid solutions, has not yet been studied. Further, this may hinder the development of hybrid ceramic materials with improved properties. Out of many possible types of complex oxides with a hollandite-like structure, one of the most promising is this one based on the $K_2O$–$TiO_2$–$Fe_2O_3$ system, in which we can expect an increase in permittivity due to increased polarization related to variable valence of the iron ions. This system was obtained earlier in works [37,38], but, until now, its dielectric properties have not been studied.

In this work, we focus on investigation of low cost and environment friendly hollandites of the $K_xFe_yTi_{8-y}O_{16}$ (KFTO) composition. The novelty of this research is due to a study of structural features and electrical properties of this group of hollandites characterized with different contents of potassium and iron.

## 2. Materials and Methods

$K_xFe_yTi_{8-y}O_{16}$ powders were obtained using a modified sol–gel synthesis according to the method described in [37]. The amount of $K^+$ and $Fe^{3+}$ in the $K_xFe_yTi_{8-y}O_{16}$ structure is usually varied in the ranges of: x = 1.4–1.8, y = 1.2–1.6; in our case, the solid solutions corresponding to the compositions of $K_{1.4}Fe_{1.6}Ti_{6.4}O_{16}$, $K_{1.6}Fe_{1.2}Ti_{6.8}O_{16}$, $K_{1.6}Fe_{1.4}Ti_{6.6}O_{16}$, $K_{1.6}Fe_{1.6}Ti_{6.4}O_{16}$, and $K_{1.8}Fe_{1.6}Ti_{6.4}O_{16}$ were synthesized (here and further are marked as KFTO-1, KFTO-2, KFTO-3, KFTO-4, and KFTO-5, respectively). The following reagents were used as sources of corresponding metals: $KNO_3$ (purity of 98%, Buyskiy himicheskiy zavod, Buy, Russia), $Fe(NO_3)_2 \cdot 9H_2O$ (purity of 98%, Buyskiy himicheskiy zavod, Buy, Russia), and titanium isopropoxide ($C_{16}H_{36}O_4Ti$, purity of 97%, Acros Organics, Geel, Belgium). Citric acid ($C_6H_8O_7 \cdot 2H_2O$, purity of 99.5%, Aricon, Moskow, Russia) was applied as a chelating agent. Ethylene glycol ($C_2H_6O_2$, purity of 98.5%, Aricon, Moskow, Russia) was used to form an organic ester. At the first stage of synthesis in the reactor, an aqueous solution containing citric acid and ethylene glycol was added to a weighed mix-

ture of potassium nitrate, iron nitrate, and titanium isopropoxide, taken in stoichiometric ratios. Further nitric acid ($HNO_3$, 65% aqueous solution, Buyskiy himicheskiy zavod, Buy, Russia) was added in the obtained mixture until complete dissolution of all the components. Citric acid and ethylene glycol were introduced in the appropriate proportions of 5 and 1.5 mol for each mol of titanium ions. Then, the 25% aqueous solution of ammonia ($NH_4OH$) was added into the mixture until pH = 8. The resulting solution was thermally treated at 250 °C to obtain a viscous polymer product, transformed into an amorphous powder as a result of auto-ignition. The resulting precursor material was grinded and calcined in air at 900 °C for 2 h. As a result, KFTO powders with different contents of potassium and titanium were obtained.

Next, 5 wt.% polyvinyl alcohol in the form of a 10% aqueous solution was added to fine KFTO particles and the homogenized powder was used to produce green body ceramics in the form of disks with a diameter of 12 mm and a thickness of 1.5 ± 0.1 mm. These disks further were sintered in air at 1080 °C for 3 h and then cooled to room temperature in a furnace. The heating rate was 190 °C/h. To study the electrical properties and phase composition, the sintered ceramic specimens were polished. The bases of the ceramic disks were coated with a silver paste and burned at the temperature of 690 °C for 1 h to form the electrodes.

The phase composition of the obtained products was identified by X-ray diffractometry using Cu(K$\alpha$) radiation ($\lambda$ = 0.15412 nm) (Thermo Scientific ARL X'TRA, Ecublens, Switzerland). A semi-quantitative phase analysis was carried out by the Rietveld method using the GSAS and EXPGUI software packages. The chemical composition of the KFTO powders was studied by BRA-135F spectrometer (BOUREVESTNIK, Saint Petersburg, Russia). The morphology of KFTO particles was analyzed by scanning electron microscopy (Aspex EXplorer, Aspex LLC, Framingham, MA, USA). The dielectric characteristics of the obtained ceramics were measured using Novocontrol Alpha AN impedance spectroscopy (Novocontrol Technologies GmbH & Co. KG, Montabaur, Germany) in the frequency range from $10^{-2}$ to $10^6$ Hz at voltage amplitude of 100 mV. The data on conductivity, permittivity, and dielectric losses were found from the experimental values of the real and imaginary parts of impedance (Z' and Z'') using well-known standard computational operations.

## 3. Results and Discussion

Figure 1 (left) shows X-ray diffraction patterns of the synthesized KFTO powders. As can be seen, X-ray diffraction patterns of all the powders, except the KFTO-2 sample, only contain the reflections corresponding to the priderite phase $K_xTi_{8-y}Fe_yO_{16}$ with a tetragonal hollandite-like structure (space group l4/m, JCPDS No 77-0990). For the KFTO-2 sample, which is characterized by the lowest iron content, the X-ray diffractogram, in addition to the main priderite, also has weak reflections of the secondary phases identified as $TiO_2$ and $Fe_2TiO_5$. It can be concluded that, in this case, monophasic $K_xTi_{8-y}Fe_yO_{16}$ powders can be only obtained when y $\geq$ 1.4. A Rietveld refinement of the XRD data was carried out to determine values of the unit cell parameters (Table 1), as well as the semi-quantitative phase composition of the KFTO-2 sample and theoretical density of the obtained powders. The XRF method made it possible to determine the quantitative ratio of K:Fe:Ti in the obtained powders, and, in combination with the XRD method, identify their chemical formulas, which are reported in Table 1. It is seen that the chemical composition of the obtained powders differs somewhat from the theoretical composition, which makes it difficult to identify the dependences of the unit cell parameters on contents of potassium and iron. However, it can be observed that the parameters of the unit cells change slightly with a change in the chemical composition. The Rietveld method made it possible to show that the KFTO-2 sample contains 11 wt. % $TiO_2$ and 3.7 wt. % $Fe_2TiO_5$, however, in this case, the potassium content of KFTO is too high (x > 2). This suggests that potassium compounds are contained in the sample in the form of amorphous phases, which cannot be identified using this method.

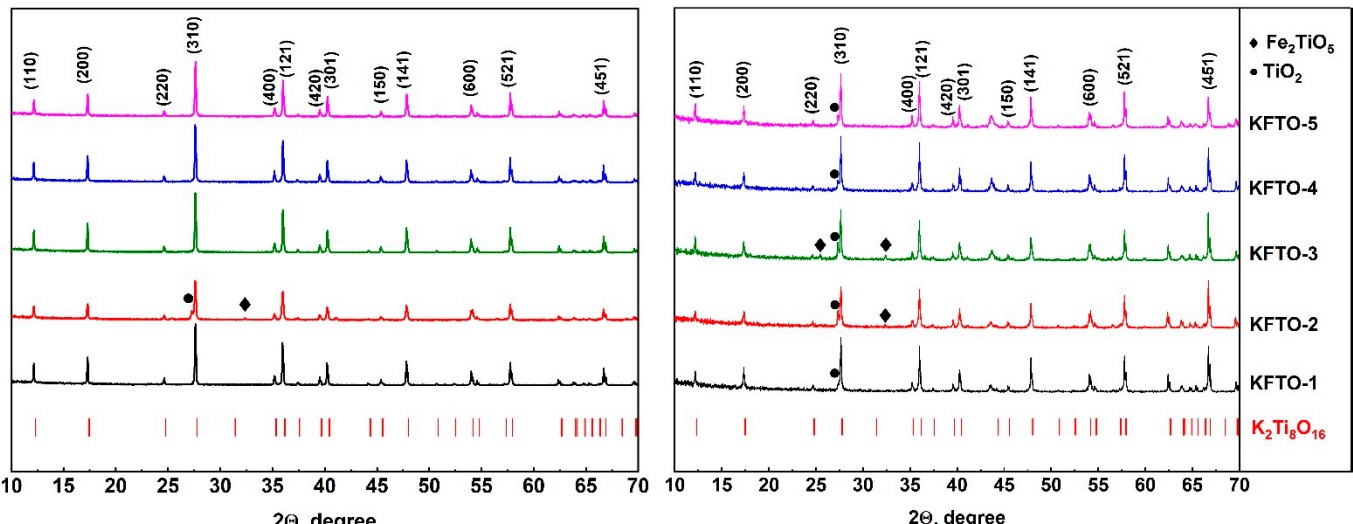

**Figure 1.** XRD patterns of the KFTO powders (**left**) and ceramics based thereon (**right**).

**Table 1.** Unit cell parameters and quantitative phase composition of KFTO powders.

| Sample | Unit Cell Parameters | | Molar Ratio K:Fe:Ti | Quantitative Phase Composition | Content of the Secondary Phases of Ceramics, wt. % |
|---|---|---|---|---|---|
| | a = b, Å | c, Å | | | |
| KFTO-1 | 10.157413 | 2.969236 | 1:1.128:4.239 | $K_{1.49}Fe_{1.68}Ti_{6.32}O_{16}$ | $TiO_2$–8.1 |
| KFTO-2 | 10.158443 | 2.967388 | 1:0.762:3.292 | $K_{2.64}Fe_{1.86}Ti_{6.14}O_{16}$ $TiO_2$—19.6 wt. % $Fe_2TiO_5$—3.7 wt. % | $TiO_2$–17.2 $Fe_2TiO_5$–2.8 |
| KFTO-3 | 10.159733 | 2.967588 | 1:0.903:4.119 | $K_{1.59}Fe_{1.44}Ti_{6.56}O_{16}$ | $TiO_2$–18.2 $Fe_2TiO_5$–3.6 |
| KFTO-4 | 10.153814 | 2.970212 | 1:1.005:3.287 | $K_{1.86}Fe_{1.87}Ti_{6.13}O_{16}$ | $TiO_2$–14.4 |
| KFTO-5 | 10.157861 | 2.967388 | 1:0.929:3.476 | $K_{1.81}Fe_{1.69}Ti_{6.31}O_{16}$ | $TiO_2$–17.7 |

XRD patterns of the KFTO ceramics of various chemical compositions sintered at 1080 °C are shown in Figure 1 (right). It is seen that, in addition to the reflections of the hollandite-like crystalline phase, some reflections of the secondary phases of $Fe_2TiO_5$ and $TiO_2$ take place in the diffractograms. As mentioned earlier, the diffraction patterns were obtained from polished ceramic samples, therefore, it can be assumed that during the sintering of ceramics, the hollandite-like structure decomposes with the formation of secondary phases $Fe_2TiO_5$ and $TiO_2$ at the grain boundaries. Depending on the chemical composition of the powders, the following features can be observed during the sintering. In the samples KFTO-1, KFTO-4, and KFTO-5 at a fixed value of y = 1.6 and variable contents of potassium (x = 1.4–1.8), only a presence of the $TiO_2$ phase is characteristic, and the intensity of its reflections increases with increasing x. For KFTO-3 and KFTO-4, it can be seen that at a fixed content of potassium (x = 1.6), a decrease in the amount of iron from y = 1.6 to y = 1.4 leads to an appearance of the reflections identified as $Fe_2TiO_5$.

Table 2 shows the values of the theoretical and measured ceramics density depending on the chemical composition. The theoretical values of density were determined by the Rietveld method using the XRD data. As can be seen, these values are in the range from 3.854 to 3.856 g/cm$^3$, which is associated with a small range of variation in the amount of potassium and iron in the structure of different KFTO samples. In addition, comparing the samples KFTO-1, KFTO-4, and KFTO-5, it can be seen that the theoretical density increases with an increased amount of potassium, and, comparing with the samples KFTO-2, KFTO-3, and KFTO-4, the density increases with [Fe] in the KFTO structure, more likely due to a larger ionic radius of $Fe^{3+}_{6/2}$ in comparison with $Ti^{4+}_{6/2}$ (0.79 and 0.75 Å, respectively).

**Table 2.** The theoretical and measured ceramics density.

|  | KFTO-1 | KFTO-2 | KFTO-3 | KFTO-4 | KFTO-5 |
|---|---|---|---|---|---|
| $P_{meas.}$, g/cm$^3$ | 3.469 | 3.433 | 3.401 | 3.300 | 3.421 |
| $P_{theor.}$, g/cm$^3$ | 3.854 | 3.855 | 3.855 | 3.856 | 3.856 |

The measured values of density are varied in the range from 3.300 to 3.469 g/cm$^3$. Due to a presence of different crystalline phases (KFTO, $TiO_2$, and $Fe_2TiO_5$) in the obtained ceramic materials, it is difficult to obtain accurate data on their porosity. However, neglecting a presence of secondary phases, it can be said that the porosity of the obtained ceramics is in the range from 10 to 14.

Typical SEM microphotograph of KFTO powders is shown in Figure 2a. As seen, the crystals have a shape matching the hollandite family I4/m tetragonal structure. The base width is approximately 250 nm and the height is 700 nm.

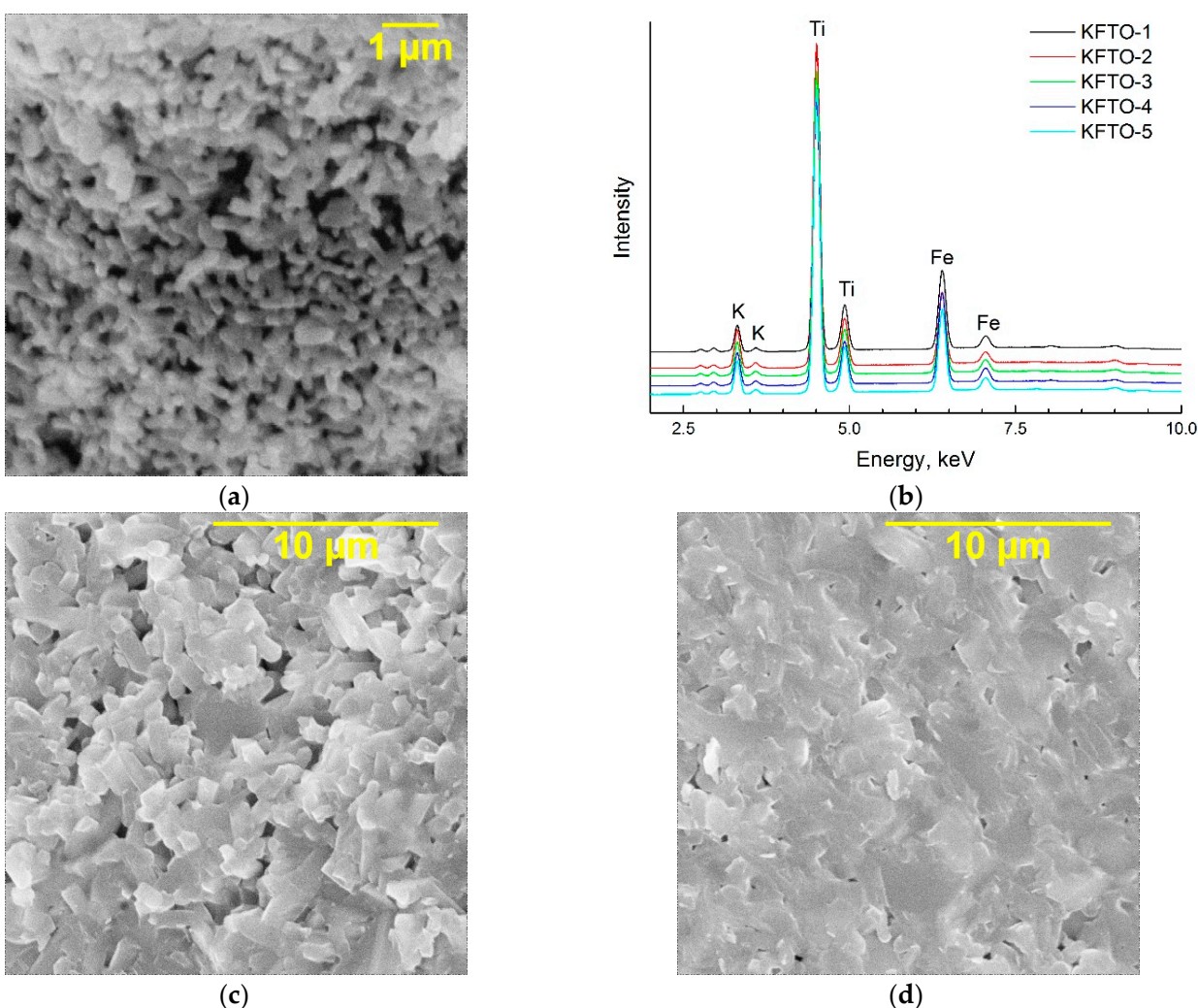

**Figure 2.** (**a**) SEM image of KFTO powder, (**b**) XRF spectrum of KFTO powder, (**c**) SEM image of the KFTO-2 ceramics, (**d**) SEM image of the KFTO-5 ceramics.

SEM micrographs of the ceramic samples KFTO-2 and KFTO-5 (cross-section) are shown in Figure 2c,d, respectively. As can be seen, in both cases, grain growth occurs. However, in the KFTO-2 sample, pronounced crystals are observed and further grain growth during the sintering is more problematic in comparison with the KFTO-5 sample, which has a microstructure with difficult-to-distinguish grain boundaries and fewer pores.

Thus, it can be assumed that the crystals of secondary phases in the parent KFTO-2 powder ($TiO_2$, $Fe_2TiO_5$, Figure 1) act as a barrier to form grain boundaries.

Frequency dependences of the KFTO ceramics permittivity in the frequency range from 0.1 Hz to 1 MHz at 25 °C are reported in Figure 3a. A value of permittivity for all the investigated KFTO ceramic compositions tends to decrease with increasing frequency, showing the phenomenon of dielectric relaxation. This behavior is typical for polar ceramic materials, since the dipoles have less time to orient themselves under the action of the electric field.

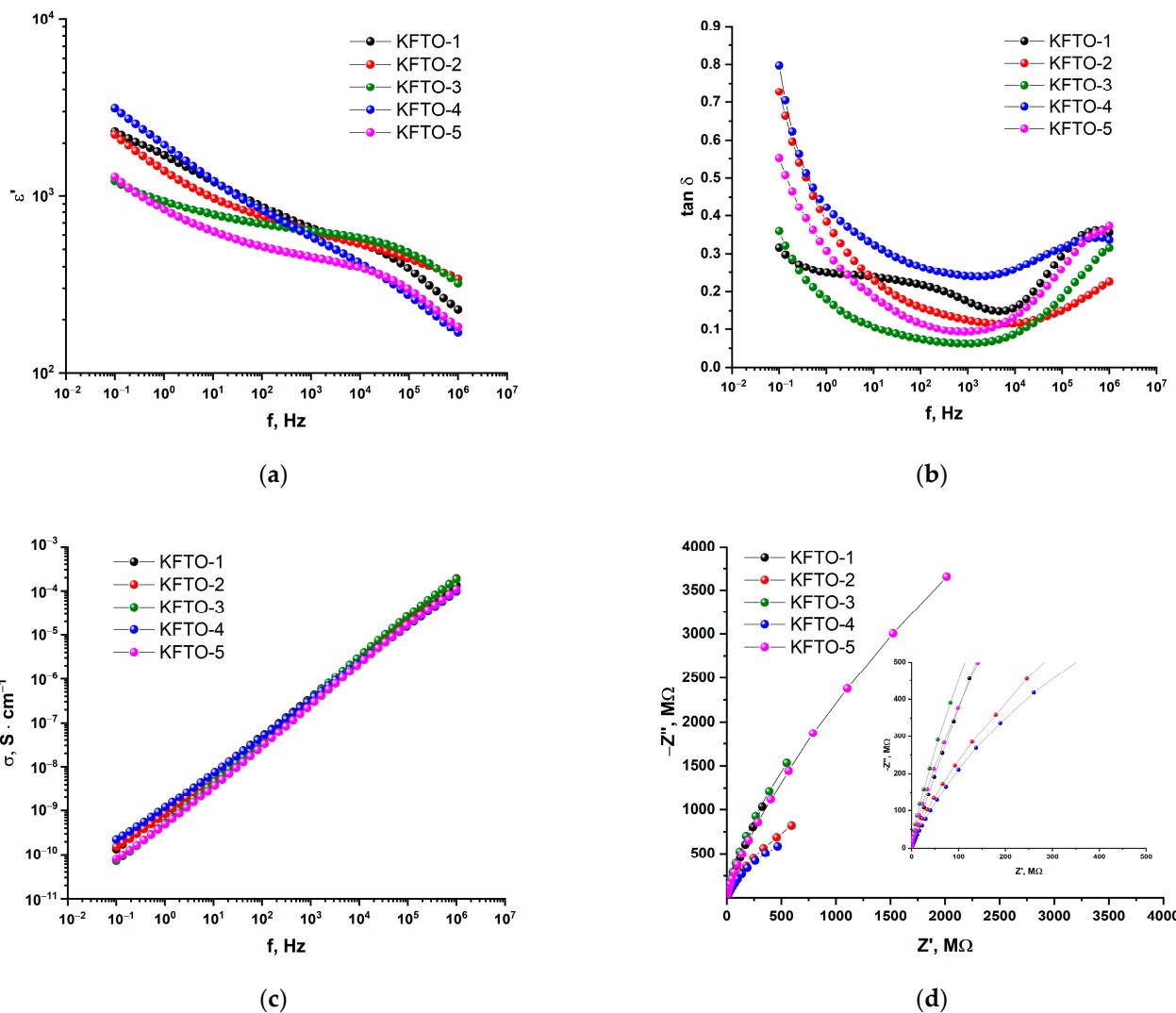

**Figure 3.** Frequency dependences of permittivity, $\varepsilon'$ (**a**); dielectric losses, *tanδ* (**b**), conductivity, $\sigma$ (**c**); and Nyquist plots (**d**) for different kinds of the KFTO ceramics.

In addition, the curve $\varepsilon'(f)$ changes the angles of inclination at frequencies of about 10 and $10^5$ Hz. This fact indicates a contribution of several types of relaxation processes to the permittivity. In addition, analyzing the graphs $\varepsilon'(f)$ for the ceramics based on KFTO-1, KFTO-4, and KFTO-5 powders, it can be seen that the permittivity of ceramics decreases with increasing the potassium content. The ceramic samples KFTO-1, KFTO-4, and KFTO-5 are characterized with $\varepsilon'$ = 666, 600, and 456 at 1 kHz and $\varepsilon'$ = 227, 167, and 180 at 1 MHz, respectively. An increase in the permittivity of KFTO with a decrease in the content of potassium can be supported by increased polarizability of potassium appearing as a result of a larger number of vacancies on A-sites. Depending on the iron content, the samples

KFTO-2, KFTO-3, and KFTO-4 have similar values of $\varepsilon' \approx 600$ at 1 kHz, whereas, at 1 MHz this parameter decreases with an increase in [Fe] ($\varepsilon'$ = 338, 320, and 227, respectively).

The frequency dependence of the dielectric loss tangent is shown in Figure 3b. For all the KFTO ceramic samples, the highest values of dielectric losses are observed in the low-frequency region. As the frequency increases to 1–10 kHz, the dielectric losses tend to decrease and reach their minimum values, followed by a sharp increase with increasing frequency. This phenomenon can be explained by the Debye dipole relaxation process. KFTO-4 has the highest $tan\delta$ = 0.24 at 1 kHz, which can be associated with the highest porosity (Table 2). For other samples, the dielectric losses decrease at 1 kHz with an increase in the amount of the $TiO_2$ phase formed in the bulk of the ceramics by sintering (Figure 1b). For the KFTO-1 ceramics characterized with a low $TiO_2$ content, the dielectric loss tangent is of 0.17, and for the KFTO-3 and KFTO-5 samples with high contents of the $TiO_2$ phase, the dielectric loss decreases down to 0.09 and 0.06 at 1 kHz, respectively. This decrease in dielectric loss can be explained by a presence of $TiO_2$, which contributes to an increase in the grain boundary insulation resistance.

As shown in Figure 3c, the real components of the AC conductivity $\sigma_{AC}\prime$ tend to increase with increasing frequency of the external field, which is similar to the cases of other potassium titanates [39] and can be explained by the universal Johnscher law:

$$\sigma_{AC}\prime(\omega) = \sigma_{DC} + A\omega^n \qquad (1)$$

where $\sigma_{DC}$ is the DC conductivity, $\omega$ is the angular frequency, $A$ is the constant representing the polarizability power, and $n$ is the exponent parameter varied from 0 to 1. Accordingly, the DC conductivity has values varying from $0.81 \times 10^{-9}$ to $2.23 \times 10^{-9}$ S $\times$ cm$^{-1}$ at 0.1 Hz, and from $1.00 \times 10^{-4}$ to $1.92 \times 10^{-4}$ S $\times$ cm$^{-1}$ at 1 MHz.

Nyquist plots of the KFTO ceramics with various compositions are shown in Figure 3d. As can be seen, all the graphs represent one semicircle only. Similar impedance hodographs can be supplemented with an equivalent circuit, which contains two series resistors ($R_g$ and $R_{gb}$) with a constant phase element (CPE), where $R_g$ and $R_{gb}$ determine the grain resistance and grain boundary resistance. In this case, the semi-circle implies that the grain resistance is low and the grain boundary resistance is approximately equal to the value of Z′ at low frequencies. In addition, one semicircle does not allow separating the impacts of volume and grain boundary contributions.

Cole–Cole plots for the permittivity are shown in Figure 4a. The semicircles and a straight line with a certain angle of inclination are observed in all the diagrams. These features indicate a presence of two relaxation processes, as well as the contribution of different components to the permittivity. The frequency dependences of permittivity and dielectric losses allow for estimating the contributions of three main processes to the polarization of dielectrics: induced dipole orientation of grains (electronically pinned defective dipoles (EPDD)), grain boundaries (internal barrier layer of the capacitance (IBLC)) and ceramic/electrode boundary [40–42]. The main contribution of grains to the permittivity occurs at high frequencies ($f > 0.1$ MHz). The boundary contribution of the IBLC model to the permittivity appears at medium frequencies (1 Hz $< f <$ 0.1 MHz), while the electrode polarization effect appears at low frequencies ($f <$ 10 Hz). The contribution of the listed relaxations to the permittivity can be analyzed using the Havriliak–Negami (HN) equation [43], which analyzes the totality of molecular relaxations, particle interactions, phase transitions, conductivity, etc., and helps to explain the differences in permittivity and dipole relaxation times. The Havriliak–Negami function has the following form:

$$\varepsilon^* = \frac{j\sigma_{DC}}{\omega\varepsilon_0} + \left(\varepsilon_\infty + \frac{\Delta\varepsilon}{\left(1 + (j\omega\tau)^\alpha\right)^\beta}\right)_{EPDD} + \left(\varepsilon_\infty + \frac{\Delta\varepsilon}{\left(1 + (j\omega\tau)^\alpha\right)^\beta}\right)_{IBLC} \qquad (2)$$

where $\varepsilon^*$ is the complex permittivity; $\omega$ is the angular frequency; $\tau$ is the average relaxation time; $\sigma_{DC}$ is the DC conductivity; $\alpha$ is the maximum width of the relaxation time variance; and $\beta$ is the peak of the parameter asymmetry, $\Delta\varepsilon = \varepsilon_s - \varepsilon_\infty$. Here, $\varepsilon_s$ is the static permittiv-

ity at low frequencies, $\varepsilon_\infty$ is the permittivity at infinitely high frequencies. For $\alpha = 1$ and $\beta < 1$, the equation can be transformed to a Cole–Davidson function (CD); for $\alpha < 1$ and $\beta = 1$ it takes the form of a Cole–Cole function (CC), and for $\alpha = 1$ and $\beta = 1$ it converts to a form of Debye's law.

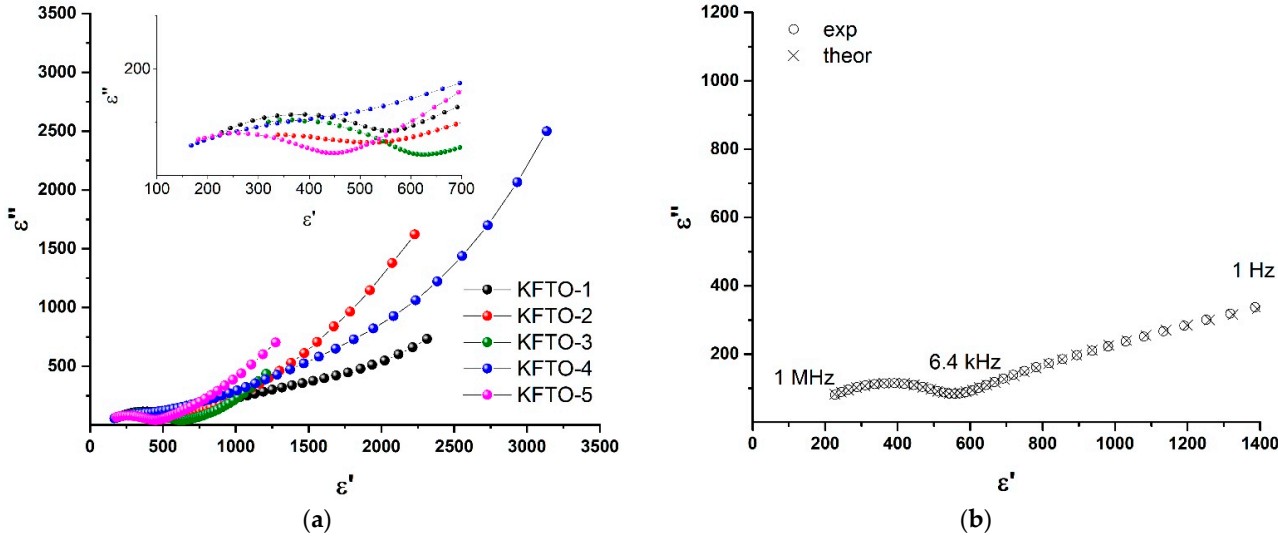

**Figure 4.** (**a**) The Cole–Cole plots of the KFTO ceramics at 25 °C, (**b**) experimental and calculated Cole–Cole plot.

The parameters of the HN equation determined using the software DielParamFit_2 [44,45], calculated with high accuracy (Figure 4b), are presented in Table 3. The calculated values of direct current conductivity are $\sigma_{DC} = 0.34$–$52.9 \times 10^{-10}$ S/cm. As shown in Table 3, the relaxation behavior of grain dipoles obeys the Cole–Cole equation ($\alpha = 0.4$–$0.9$, $\beta \approx 1$). This means that grain dipoles are represented not by a single relaxation time, but by a set of relaxation times, the distribution of which narrows as the parameter $\alpha$ increases. The calculated dipole relaxation time of the grain is varied from $1.03 \times 10^{-6}$ to $5.51 \times 10^{-6}$ s. Analyzing the parameters of the HN equation, we can see that a higher value of $\alpha$ corresponds to a narrower distribution of relaxation times. The relaxation process in the grains is caused by the migration of $K^+$ ions inside the channel to neighboring vacancies. At the same time, as the potassium content increases, the number of vacancies decreases, hence, the number of variations of overshoots to longer distances decreases and causes an increase in the parameter $\alpha$. This is observed for the KFTO-5 ceramic, which is characterized by the largest lattice parameter $c$ at high potassium content. The KFTO-1, KFTO-3, and KFTO-4 samples are characterized with a wider distribution of relaxation times, which is associated with a larger parameter $c$ and lower potassium occupancy. The KFTO-2 ceramic is characterized by the smallest size $c$ at high potassium content, which corresponds to a high value of the parameter $\alpha$.

**Table 3.** Results of the Havriliak–Negami function fitting from the experimental data using Diel-ParamFit_2 software.

| Parameters | KFTO-1 | | KFTO-2 | | KFTO-3 | | KFTO-4 | | KFTO-5 | |
|---|---|---|---|---|---|---|---|---|---|---|
| | EPDD | IBLC | EPDD | IBLC | EPDD | IBLC | EPDD | IBLC | EPDD | IBLC |
| $\Delta\varepsilon$ | 451 | 1134 | 377 | 511 | 306 | 2119 | 403 | 2734 | 554 | 527 |
| $\alpha$ | 0.37 | 1 | 0.51 | 0.88 | 0.41 | 1 | 0.31 | 0.92 | 0.58 | 1 |
| $\beta$ | 1 | 0.24 | 0.77 | 0.35 | 1 | 0.25 | 0.26 | 1 | 0.44 | 0.21 |
| $\tau$,s | $5.51 \times 10^{-6}$ | 0.356 | $1.15 \times 10^{-6}$ | 0.197 | $2.53 \times 10^{-6}$ | 0.385 | $3.74 \times 10^{-6}$ | 0.259 | $1.03 \times 10^{-6}$ | 0.689 |
| $\sigma_{DC}$, S·cm$^{-1}$ | $1.63 \times 10^{-10}$ | | $0.82 \times 10^{-10}$ | | $1.89 \times 10^{-10}$ | | $52.9 \times 10^{-10}$ | | $0.34 \times 10^{-10}$ | |

For dipoles at the grain boundary, the parameters of the HN equation are $\alpha \approx 1$ and $\beta = 0.24$–$0.35$. This means that the grain boundaries have the population of interacting dipoles characterized with dispersion of the relaxation times and a shift in the distribution to the high-frequency region. In this case, small values of $\beta$ (<0.5) promote an increase in permittivity at higher frequencies and the contribution of grain boundary dipoles to the permittivity is observed at $f$ >1. The calculated average relaxation time of the accumulated dipoles at the grain boundary is varied from 0.197 to 0.687 s. A similar dependence is observed for the relaxation time. For the samples KFTO-2 and KFTO-5, relaxation times have the smallest values. The analysis of the HN equation parameters for IBLC does not allow us to identify a characteristic trend in the dependence on chemical composition of the KFTO ceramics, due to the complex formed grain boundary. In addition, the calculated values of permittivity ($\Delta\varepsilon$) for IBLC are much higher than those estimated for EPDD. Thus, we can conclude that, in the high-frequency region, the permittivity of the KFTO ceramics is determined mainly by polarization of the dipoles of grains, and, as the frequency decreases, the polarization of the dipoles at the grain boundary contributes most to the permittivity.

## 4. Conclusions

Some solid solutions with a hollandite-like structure and $K_xFe_yTi_{8-y}O_{16}$ chemical composition (KFTO) were successfully synthesized using a modified sol–gel technique. The XRD and SEM data show that the obtained mono-phase KFTO powders have a tetragonal structure at x = 1.4–1.8 and y = 1.4–1.6. The ceramic materials based on KFTO powders were produced by compressing and sintering at 1080 °C for 3 h. It was recognized that the hollandite-like crystalline phase could be partially broken during the high-temperature treatment, forming $TiO_2$ and $Fe_2TiO_5$ crystals distributed over the volume of the ceramic material. Frequency dependences of the electric properties were studied using impedance spectroscopy. It is found that increased contents of the rutile secondary phase promote a decrease in dielectric losses during the sintering, whereas reduced concentrations of potassium in the KFTO composition lead to an increase in permittivity. In addition, an appearance of the $Fe_2TiO_5$ phase in the obtained ceramics prevents the sintering process. The contribution of electron-pinned defect dipoles (EPDD) and internal barrier layer capacitance (IBLC) to the permittivity were estimated using the Havriliak–Negami equation. In accordance with the calculated parameters of the Havriliak–Negami equation, the complex nature of the dielectric relaxation processes taking place in the KFTO-based ceramics was shown. The dipole relaxation times of the grain and grain boundary were estimated as varying in the range from $1.03 \times 10^{-6}$ to $5.51 \times 10^{-6}$ s and from 0.197 to 0.687, respectively.

**Author Contributions:** Conceptualization, A.T. and N.G.; methodology, A.T.; software, D.A.; validation, M.V.; formal analysis, A.T. and D.A.; investigation, N.M., I.Z. and S.B.; resources, N.G.; data curation, A.G.; writing—original draft preparation, A.T.; writing—review and editing, D.A. and M.V.; visualization, D.A. and N.G.; supervision, D.Z. and A.A.; project administration and funding acquisition, N.G.; writing—review and editing, A.G. All authors have read and agreed to the published version of the manuscript.

**Funding:** This research was funded by the Russian Science Foundation, grant number 19-73-10133, https://rscf.ru/en/project/19-73-10133/ (accessed on 3 August 2022). The funders had no role in the design of the study; in the collection, analyses, or interpretation of data; in the writing of the manuscript; or in the decision to publish the results.

**Institutional Review Board Statement:** Not applicable.

**Informed Consent Statement:** Not applicable.

**Data Availability Statement:** Not applicable.

**Conflicts of Interest:** The authors declare no conflict of interest.

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
