# Peer review of "Synthesis and Dielectric Relaxation Studies of KxFeyTi8-yO16 (x = 1.4–1.8 and y = 1.4–1.6) Ceramics with Hollandite Structure"

_ceramics, doi:10.3390/ceramics6010037_

Round 1

Reviewer 1 Report

1.       Already above 100k maximum permittivity has been reported before many times by improving the CCTO crystal quality or in the compactly packed granular and smaller porosity CCTO ceramics. Moreover, non-Debye relaxation is also reported in the core-shell titania/CCTO composites {Journal of Materials Science: Materials in Electronics volume 33, pages9395–9402 (2022)}. It is difficult to understand the choice of the material selected (KFTO) in the manuscript provided that the better options of core-shell geometries are also available. In a similar note, authors are also reporting smaller permittivity values that too without accounting for contemporary developments, which again creates doubts, such as, what’s good in the selected KFTO material? Hence the novelty in the submitted manuscript is not suitable for consideration in Ceramics-MDPI, unless it is explicitly mentioned in the introduction part of the submitted manuscript.

2.     Title of the manuscript is unclear and lacks complete information. For instance, authors are not measuring a complete range of electric properties, just a measurement of ac-conductivity is presented in the results section. I would suggest some other title like “Studies of ac-conductivity and dielectric properties of KFTO ceramics”, or any other title that is based explicitly on the results presented.

3.      The fact that only a single composition of KFTO is considered in the study presented should be clearly mentioned in the abstract. Rather, the current form of abstract gives an impression of a range of compositionally vivid samples of KFTO which have been synthesized in this work. Everywhere in the abstract, the plural wordings such as ceramics, “have been” are used for the samples, and should corrected to singular wordings.

4.       Just by considering only a single sample of KFTO, I think the study is not complete, since any change in the potassium concentration (x) should affect the dielectric relaxation via interfacial charge accumulation. Effects of (x) are completely ignored in the discussion section. I suggest at least one more sample should be studied at a different x value.

5.       On line 125, authors are stating that the KFTO semiconducting grains are separated by insulating grain boundaries. The SEM image (Figure 2) presented for this is not clear enough to see the grain boundaries and insufficient to prove their statement. I suggest authors should perform the HRTEM studies to look clearly into the interfaces at a higher resolution. Moreover, the microscopic section is very short and needs more explanation in the manuscript.

6.       Authors are suggesting a possible loss of potassium at a much lower temperature than the phase transition temperature of common potassium titanates 1500°C ,which is difficult to apprehend without a through compositional analysis such as XPS or EDX results. Conclusion of the manuscript either be changed accordingly, or if not, then clearly mention how much percentage change in K-concentration taking place on annealing.

Reviewer 2 Report

This paper is not suitable for publication.

The major defects of the manuscript are:

1. Lines 103-105: “... as a result of oxygen loss ..., subsequent re-oxidation...” Does any evidence support this hypothesis? Oxygen loss would result in reduction of the materials, e.g., decrease in the valency of cation.

2. The XRD pattern of the sintered ceramic shows the signals of second phases. It is well known that the phase assemblage at as-sintered surface and bulk region may be different for sintered ceramics. Which type of sample was used for the XRD analysis (as-sintered surface, polished surface, or powder)?

3. Dielectric properties of a sintered ceramic are remarkably affected by its microstructure. The porosity data of the sintered ceramic should be provided. In Microscopy section, only the as-calcined powder was examined by SEM. It is strongly suggested that the microstructure of the sintered ceramic, using as-sintered surface a/o polish-etched surface, should be carefully examined. What are the pore morphology, grain size, and second phase formation (to compare with the XRD result)?

4. In the impedance section, the authors state “... the microstructure of KFTO(H) ceramic is formed by semiconducting grains separated by insulating grain boundaries.” Does any grain boundary phase or solute segregation can be found under microstructural examination?

5. Lines 133-134: “...0 n 1.” and “... 4.5 ´ 10-9...” should be corrected.

6. Conclusions Section:

Line 186: “the hollandite phase on the grain surface can destroy.......as a result of the loss of potassium ...” What is meant by “grain surface”? Is that mean grain boundary or the external sample surface? Meanwhile, “loss of potassium” is not mentioned in the Results and discussion section. What is the data supporting this conclusion?

Minor points

1. The following symbols should be defined at first mention and used consistently thereafter.

     K (line 30), e¢ (line 33) and e (line 46), tand (line 46).

The abstract should not contain any undefined abbreviations.

2. Line 28: What is meant by “high dielectric properties”? Dielectric properties contain many parameters, e.g., dielectric constant, loss tangent, dielectric field strength, etc.

3. Line 120: “... the increasing alternating field.” should be “... the increasing frequency of alternating field.”

Reviewer 3 Report

In this manuscript, the author synthesized Fe-doped potassium titanate via a sol-gel method and measured the frequency response of its dielectric properties. However, lack of novelty and detailed explanation to the experimental observation hinder the publication of this article. The author states that the K2O-TiO2-Fe2O3 system is promising due to high permittivity introduced by iron doping. It could be very interesting to see the effect of different amounts of iron doping on the dielectric properties. Moreover, the origin of the polarization relaxation with two different relaxation times has not been clearly explained. The frequency dependence of the conductivity needs more discussion. The author states the generation of the TiO2 and Fe2TiO5 phases is due to oxygen loss. Evidence is also needed.

Round 2

Reviewer 1 Report

I find the current form of the manuscript entitled “Synthesis and Dielectric Relaxation Studies of KFTO Ceramics with Hollandite Structure” is suitable for publication. The manuscript is based on the sol-gel synthesis of Fe-doped potassium titanate with several different chemical compositions and characterized structurally and by impedance spectroscopy. Authors have successfully lower down the dielectric losses via thermal degradation of  Hollandite phase owing to the formation of several secondary phases.    

Author Response

Thank you for your comment!

Reviewer 2 Report

The revised manuscript is improved.

Extensive editing of English language is required.

Author Response

Thank you for your comment! Extensive editing of English language was done.

Reviewer 3 Report

1. The author states that the formation of secondary phases results from the decomposition of the the hollandite-like structure at the grain boundaries due to polish. It could be better proved by comparing the XRD patterns of unpolished samples. 

2. The quantitative phase composition of the ceramic samples is suggested to be calculated by the Rietveld refinement.

3. A detailed explanation on the relation between the parameters extracted from the Gavrilyak-Negami function and the formulation of the samples is needed. How do the cation variation, phase composition and microstructure influence the relaxation mechanism?

4. The manuscript needs to be carefully proofread and the improvement in written English is also needed.

Author Response

Thank you for your comments!

  1. The description of the sample preparation mode and comparison parameters of X-ray diffraction patterns of powders and sintered pellets has been improved in the Materials and Methods section, as well as in the discussion of figures and tables. On the surface of unpolished samples, the formed phases are present in a larger amount, however, the analysis of polished surfaces is used.
  2. The quantitative phase composition of the ceramic samples was calculated by the Rietveld refinement (Table 1).
  3. A detailed explanation on the relation between the parameters extracted from the Havrilyak-Negami function and the formulation of the samples was added.
  4. The improvement in written English was done.

Round 3

Reviewer 3 Report

The author has addressed all my questions. I have no more comments.